# Division and Analysis of Accident-Prone Areas near Highway Ramps Based on Spatial Autocorrelation

**Qing Ye** [1,2,3,†], **Yi Li** [4,*,†], **Wenzhe Shen** [4] **and Zhaoze Xuan** [4]

1. National Engineering and Research Center for Mountainous Highways, Chongqing 400067, China; yeqing2@cmhk.com
2. Research and Development Center of Transport Industry of Self-Driving Technology, China Merchants Chongqing Communications Technology Research and Design Institute Co., Ltd., Chongqing 400067, China
3. School of Big Data & Software Engineering, Chongqing University, Chongqing 400030, China
4. Logistics Research Center, Shanghai Maritime University, Shanghai 201306, China; 202130510078@stu.shmtu.edu.cn (W.S.); 202130510119@stu.shmtu.edu.cn (Z.X.)
* Correspondence: liyi205598@shmtu.edu.cn
† These authors contributed equally to this study.

**Abstract:** This study focuses on identifying accident-prone areas and analyzing the factors contributing to the distribution of traffic accidents near highway ramps. A combined method of kernel density estimation, spatial autocorrelation analysis, and multivariate logistic regression analysis helped to identify accident hotspots. Through data collection and analysis, the clustering characteristics of traffic accidents in the diversion and merging areas were identified. Four levels of accident-prone areas were divided according to the accident rates. The factors influencing the spatial distribution of accidents were analyzed. The results showed that traffic accidents in the diversion area were concentrated near the exit, but the accidents in merging areas had a wider range of distribution. The analysis of this phenomenon was conducted using the multinomial logit model results. The important factors of different accident-prone areas were clarified. The temperature, the accident lane, weather conditions, and the time of day had significant impacts on the spatial distribution of traffic accidents. The study's findings provide an important decision-making basis for highway accident prevention management.

**Keywords:** diversion area; merging area; spatial autocorrelation; kernel density method; multivariate logit regression

## 1. Introduction

With the rapid development of automobiles and autonomous vehicle technologies, how to decrease traffic accidents is increasingly receiving attention in various countries. In the future, human-driven vehicles may gradually be replaced by autonomous vehicles, but during this process, the road traffic will be in a mixed state of autonomous and human-driven vehicles. Vehicle conflicts may arise due to the perception, communication, and speed differences of autonomous vehicles in mixed traffic flows in highway ramp areas. The traffic conditions in the diversion and merging areas, which serve as the connection points between the arterial roads and highways, are extremely complex and prone to traffic accidents. According to data provided by the World Health Organization, about 1.35 million people die each year in road traffic accidents worldwide, equivalent to 3700 every day globally [1]. Although the occurrence of traffic accidents is random, the temporal and spatial distribution of accidents on the specific road segments show some characteristics in different road conditions and traffic environments. The special sections of highways (such as merging and diversion areas) are black-spots for traffic accidents, especially for major accidents. Therefore, the corresponding accident analysis of these areas has gradually become a research topic in recent years [2–5]. Especially in recent years,

machine learning algorithms have been widely used to divide and analyze accident-prone areas [6–8].

The merging and diversion areas of highways are important highway nodes in the transportation system. They are also areas where traffic accidents frequently occur. The vehicles going straight in the merging and diversion areas of highways often travel fast, while the vehicles entering or exiting the highway need to slow down. The large speed difference can easily cause traffic accidents. Past studies have shown that merging and diversion behaviors are among the main causes of accidents. Compared with general road segments, the spatial competition ratio in merging and diversion areas is higher. Rear-end and side-collision accidents are more likely to occur in such areas [9,10]. In order to explore the causes of traffic accidents, researchers have proposed many traffic safety analysis methods to reduce or eliminate traffic accidents [11,12]. Identifying the black-spots of highways is an important method to understand and predict accidents. There are several statistical methods for identifying accident-prone areas. In order to explore the impact of risk factors on the collision frequency and severity of large trucks, Dong et al. [13] applied multinomial logit (MNL) and negative binomial (NB) models to analyze collision severity and frequency, respectively. The results showed that the driver's age, speed limit, and location type only had significant effects on the frequency of large truck crashes. Pulugurtha et al. [14] used the geographic information system (GIS) method to study the spatial patterns of pedestrian collisions in order to identify the areas with high incidences of pedestrian collisions. The results indicated that there were significant differences in the ranking results when each method was considered. Dellinger et al. [15] explored the factors that influence fatal crashes among U.S. drivers over the age of 55. The results showed that the number of fatal crashes increased with age. Meanwhile, the relative contributions of collision incidence density and exposure prevalence were greater than collision mortality. To improve road safety, Pritee et al. [16] used geographic information systems for risk assessment and statistical analysis to identify high-frequency accident sites. The results showed that it was beneficial to divide the accident site by using the heat map analysis method of kernel density estimation. Jahan et al. [17] developed a hybrid method, based on accident type, for improving the rate quality control method to overcome the shortcomings. According to the case study and based on the results in the real environment, the proposed method could detect and identify the accident hotspots. Ray et al. [18] presented a new method to rank the severity of roadside hazards based on observable crash data. Unlike the earlier subjective severity index method, the new EFCCR method was based on observed crash data and used a systematic approach to calculate crash severities. Zou et al. [19] proposed methods based on the Ordered Weighted Averaging (OWA) operator and Uncertain Ordered Weighted Averaging (UOWA) operator to fuse different certain results and different interval results to one result, respectively. The research provided more reliable choices to describe different results obtained from different methods in accident reconstruction. For the purpose of studying and comparing the effects of the methodological diversity of road network segmentation on the performance of different BSID methods, Ghadi et al. [20] evaluated four commonly applied BS methods (empirical Bayesian (EB), excess EB, accident frequency, and accident ratio) against four different segmentation methods (spatial clustering, constant length, constant traffic volume, and the standard Highway Safety Manual segmentation method). The results showed that the EB method had surpassed the other BSID methods for all segmentation approaches.

In past studies, researchers have evaluated traffic accidents in various cities and rural areas. In general, these studies can be divided into two categories. The first is (a) analyzing the factors that cause traffic accidents [21–23]. These studies tried to determine the interactions among environmental, human, road, and vehicle characteristics and traffic accidents. The second is (b) utilizing various geospatial analysis methods to map black-spots of traffic accidents. For example, the kernel density estimation (KDE) method was commonly used to illustrate the density of traffic accidents based on the number of accident points at each spatial location [24,25]. In addition, some studies used distance-based

methods to determine the clustering of traffic accidents [26,27]. Some studies used time series analysis to explore the evolution of traffic accident clusters [28,29]. Time series analysis and spatial clustering analysis were combined to identify accident-prone locations; for example, Kingham et al. studied the temporal evolution and spatial clustering of traffic accidents in Christchurch, New Zealand [30]. Liu et al. [31] studied the temporal evolution of fatal accidents in Iowa (USA) and found that fatal car accidents in all Iowa counties declined from 2006 to 2015, but the rate of decline varied across counties.

In recent years, with the development of artificial intelligence and machine learning, some emerging algorithms have been used to identify accident-prone locations. For example, Meng et al. [32] established a self-organizing neural network model for identifying accident-prone locations. They proposed a process for identifying prominent accident-inducing factors based on the combination of discrete multivariate algorithms and probability distributions. Wang et al. [33] applied the DENCLUE clustering algorithm in identifying accident-prone locations, which can effectively avoid the pre-division of investigation locations and achieve arbitrary length clustering compared to traditional methods. Ifthikar et al. [34] identified accident-prone areas and related causes by clustering accident location coordinates. Qiu et al. [35] proposed an improved DBSCAN clustering algorithm to identify traffic-accident-prone areas by selecting reasonable values for the parameters $\varepsilon$ and minPts. Yakar et al. [36] studied the application of the relative frequency method in determining accident-prone road sections. Zhao et al. [37] developed deep convolutional embedded clustering (DCEC) to classify traffic flow into nine states. The results of the logistic regression model proved that the nine traffic states were significantly associated with crash risk in the vicinities of weaving segments, and each traffic state could be assigned a unique safety level.

However, most of these methods were based on statistical analysis and only focus on non-spatial features and attributes of the data. Geographical spatial properties were not associated with accident occurrence. Traditional statistical analysis usually considered the occurrence of traffic accidents as a random and independent process. Corresponding geographical spatial information of these accidents was often omitted. However, the spatial data of an accident are not always the same as other accidents. Even if two accidents happened at the same location, their traffic environment and timestamp will never be the same. Therefore, an unreasonable combination of these factors will lead to information loss and unreasonable clustering results for accident-prone areas.

To solve this problem, the spatial autocorrelation-based method is an ideal way to identify the geographical location relationship of various accidents and to reflect the spatial correlation of different accident attributes. This method is also suitable for the safety evaluation of highway black-spots. For example, Khanh et al. [38] proposed a method for determining the location of traffic accident black-spots by combining the kernel density estimation (KDE) algorithm and spatial autocorrelation analysis. Fan [39] used a spatial autoregressive quantile model to estimate how risk factors affect overall and fatal traffic accident rates. The results were expected to provide strategies for reducing accident rates and improving road safety. Khaled et al. [40] used spatial autocorrelation (Global Moran I Index) and local hotspot analysis (Getis-Ord Gi*) in a GIS environment to determine the spatial patterns and temporal evolution of accident black-spots along the internal and main road networks in the study area. Fan et al. [41] trained and optimized a complex model for identifying accident black-spots using the support vector machine method based on the structured association features of urban traffic accident big data, which improved the accuracy of black-spot identification. Tanprasert et al. [42] proposed a new technology that used street-view images to identify black-spots. This technology was based on the hypothesis that the features of the surrounding road environment had an impact on the safety level of specific locations. It was the first black-spot classification technology that was fully environment-aware. Vitianingsih et al. [43] presented a framework of spatial analysis using a hybrid estimation model based on a combination of multi-criteria decision

making (MCDM) and artificial neural network (ANN) (MCDM-ANN) classification. This model is useful for traffic-accident-prone road classification with a spatial dataset.

Although the above studies have proposed various methods to identify black-spots or accident-prone areas, most results remained at the macroscopic level of road traffic. The spatial characteristics of complex traffic environments were not considered or described quantically in these studies. In high-speed road sections and special nodes, the black-spots often occurred at some specific positions, which had strong correlations on the microscopic scale. Therefore, traditional accident statistics methods are not suitable for such road sections. To solve this problem, this paper utilized the microscopic spatial autocorrelation method to divide the accident-prone areas into highway diversion and merging areas.

The rest of this study is organized as follows. Section 2 introduces the proposed method. Section 3 presents the experiments and data. Section 4 analyzes and verifies the results. The analysis and discussion are presented in Section 5. The conclusion is presented in Section 6.

## 2. Methodology

From the spatial distribution perspective, traffic accidents are not evenly distributed. In some areas, they cluster in one location, while in other regions, this phenomenon may not be present. This phenomenon also occurs near highway ramps. Therefore, geographic information system (GIS) technology was used to analyze the spatial characteristics of traffic accidents.

First, the locations of traffic accidents near the highway ramp were geocoded on the digital road network. However, considering that merging and diversion areas on highways are relatively small micro scenes, and that road traffic accident data are often obtained from macro road networks in a country or province, the object of this study was specific scenes of special sections of highways. This required classifying accident data based on the characteristics of each special section, e.g., classifying scenes based on the number of lanes on the highway. After determining the category of special section scenes, combined with the road network map of each special section, a hotspot map of the location of traffic accidents was drawn in special sections.

Next, the distribution of incident points needed to be checked to see if it matched the clustering distribution for the next cluster analysis. This required testing the random distribution of traffic accidents in a specific section of the highway, applying the average nearest neighbor method.

After confirming that accidents conform to the law of aggregation, the kernel density estimation method was applied to calculate and draw a density map of traffic accidents. Finally, in order to evaluate the distribution pattern of traffic accidents, we used the multivariate logit regression model to find the causal factors of differences in frequent accident locations and determine the explanatory variables that cause changes in the location of accidents. The flowchart of this study is shown in Figure 1.

The following sections introduce the detailed methods used in this study, including kernel density estimation, the average nearest neighbor method, and multivariate logit regression analysis.

### 2.1. Kernel Density Estimation

The core idea of kernel density estimation (KDE) is that geographic phenomena and events can occur at any location in the spatial plane, but the probability of occurrence varies by location. Areas with dense points have a higher probability of event occurrence, while sparse areas have a lower probability. Therefore, KDE is particularly useful for analyzing and displaying point data. The geometric interpretation of the kernel density is that the density distribution is highest at the center of each point $x_i$ and decreases outward. It will reach 0 at a certain threshold range (the edge of the window) from the center, as shown

in Figure 2. The kernel density at the grid center $x$ is the sum of the densities within the window range, as shown in Equation (1).

$$\hat{f}(x) = \frac{1}{nh^d} \sum_{i=1}^{n} K(\frac{x - x_i}{h})$$

(1)

where $K(\ )$ represents the kernel function, $h$ is the bandwidth, $n$ is the number of points in the study area $R$, $d$ is the dimension of the data, and $(x - x_i)$ represents the distance from the estimation point to the event point $x_i$.

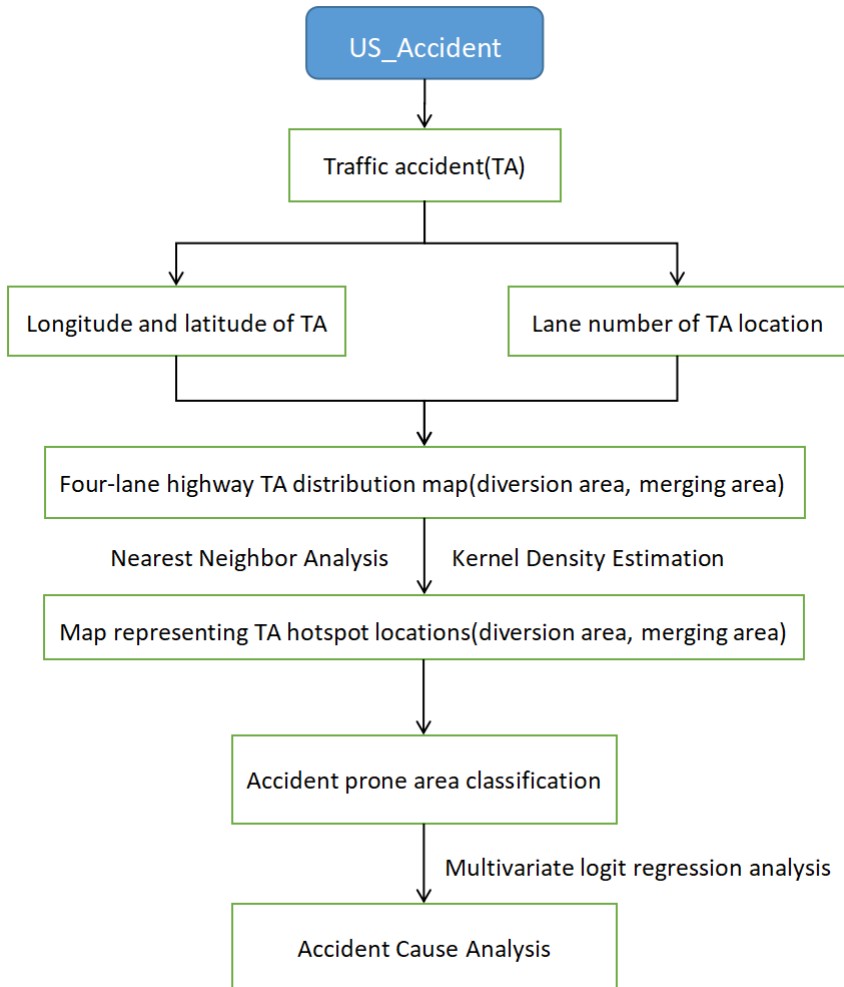

**Figure 1.** The flowchart of this study.

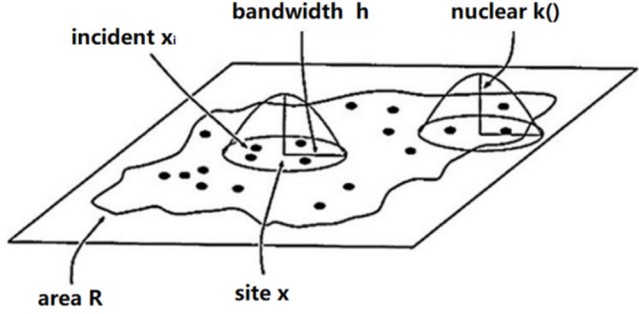

**Figure 2.** Kernel density estimation principle schematic diagram.

For example, when $d = 2$, a commonly used kernel density equation for two-dimensional plane space can be defined as shown in Equation (2):

$$\hat{f}(x) = \frac{1}{nh^2}\sum_{i=1}^{n} K(\frac{x - x_i}{h}) \tag{2}$$

Many researchers have pointed out that bandwidth is the most critical criterion for determining the most appropriate density surface. Therefore, the choice of bandwidth will significantly affect the results of hotspots. In other words, the smaller the bandwidth, the smaller the hotspots. The smoothness of the density surface is affected by the bandwidth—the smoother the density surface, the larger the bandwidth. Therefore, selecting the best bandwidth is crucial. According to the research of many scholars, the bandwidth is usually within the range of 20 to 1000 m.

*2.2. Mean Nearest Neighbor Analysis*

The Nearest Neighbor Index (NNI) is the ratio of the average observed distance between points to the expected average distance between points. If the observed distance is less than the expected distance, it indicates a clustered distribution of points. The greater the difference between the two, the stronger the clustering. If the observed distance is greater than the expected distance, it indicates a dispersed distribution of points. The formula for calculating the Nearest Neighbor Index is as shown in Equation (3).

$$N = \frac{D_o}{D_e} \tag{3}$$

where $D_e = 0.5/\sqrt{\frac{n}{A}}$, $N$ is the nearest neighbor index, $D_o$ is the observed mean distance, $D_e$ is the expected mean distance, $n$ is the number of points in the study area, and $A$ is the area of the study area.

*2.3. Multinomial Logistic Regression*

The multinomial logit model can be viewed as a joint estimation of multiple binary logit models formed by pairing each selection category of the dependent variable. The model is specified as follows in Equation (4):

$$ln\left(\frac{P(y_i = j|x)}{P(y_i = b|x)}\right) = x_i'\beta_j \tag{4}$$

where $b$ is the reference category, $j$ is the total number of categories in the categorical variable, and $\beta_j$ is the coefficient of the $j$ categorical variable. When $j = b$, the left-hand side of the Equation (4) is $ln1 = 0$, then $\beta_b = 0$. This means that the log-odds of choosing a certain category relative to the reference category is always 0, causing any explanatory variable coefficients corresponding to this category to be 0 as well.

## 3. Data Selection and Processing

*3.1. Dataset*

The research areas of this study are the merging and diversion sections of highways in the United States. The traffic accident dataset is the US_Accidents dataset, which covers 49 states in the United States. The data were collected from several data providers from 2016 to 2021, including two APIs that provided traffic incident data captured by various entities such as the federal and state transportation departments, law enforcement agencies, traffic cameras, and traffic sensors in the road network. Currently, there are approximately 3 million accident records in this dataset.

According to recent data released by the National Highway Traffic Safety Administration (NHTSA) of the United States, the number of traffic accident fatalities in the United States in 2021 reached 42,915, which is 10.5% higher than 38,824 in 2020. It is a new record

in 16 years. Particularly, accidents near the merging and diversion sections of highways are more prominent, accounting for 50% of the total accidents on highways. The causes of the accidents include drivers being unfamiliar with the road, overly relying on navigation voice prompts, and neglecting to observe highway signs, leading to missing exits, stopping, changing lanes near exits, and ultimately causing traffic accidents. Therefore, it is important to study the distribution of accident black-spots in the merging and diversion sections of highways to figure out the patterns.

The US_Accidents dataset used in this study contains 49 fields, including accident ID identification, start and end time of the accident, start and end location of the accident, the severity of the accident, natural language description of the accident, etc. The data parameters, value ranges, and descriptions are shown in Table 1.

**Table 1.** Description of dataset parameters.

| Parameter | Parameter Range | Parameter Description |
|---|---|---|
| ID | A-1, A-2, ... | Accident record unique identifier. |
| Start_Time | 8 February 2016 0:37, ... | Displays the start time of the accident in the local time zone. |
| End_Time | 8 February 2016 6:37, ... | Displays the end time of the accident in the local time zone. |
| Start_Lat | 40.10891, ... | Displays the latitude in the GPS coordinates of the accident starting point. |
| Start_Lng | −83.09286, ... | Displays the longitude in the GPS coordinates of the accident starting point. |
| Severity | 1, 2, 3, 4 | Displays the severity of the accident, with 1 indicating the least severe and 4 indicating the most severe impact. |
| Distance | 3.23, 0.747, ... | The length of the road affected by the accident (in miles). |
| Description | - | Natural language description of the accident. |
| Junction | TRUE/FALSE | If there is a junction nearby. |

The selected data include 160 accidents in the diversion area and 100 accidents in the merging area. The statistical distributions of the parameters "Severity" and "Distance" are shown in Figures 3 and 4. It is observed obviously that most of the accidents belong to Severity 2 and the impact scope of the accidents is within 0–1 miles.

*3.2. Accident Spatial Distribution near the Highway Ramp*

According to the traffic accident database, the following characteristics were used to identify the accidents that happened in the highway merging and diversion areas: latitude and longitude, description, and junction. The selected data were marked on a standard four-lane highway diversion area and a merging area through ArcGIS, as shown in Figures 5 and 6. Since the accident points selected in this paper are all near the joint (the corresponding data field is "joint = true"), the distribution of accidents on the map is within approximately 200 m of the ramp.

As can be seen from Figure 5, in the highway off-ramp area, most historical traffic accidents occurred near the off-ramp exit, and most of them occurred in the first, second, and third lanes. This is due to the improper speed control or acceleration of vehicles on the main line when the traffic volume decreases as the off-ramp vehicles leave the vehicle group. In contrast, in Figure 6, compared with the off-ramp area, the distribution of traffic accidents in the on-ramp area is relatively discrete, and most accidents occurred in the first and second lanes on the main line near the entrance of the ramp. This means that the vehicles on the main line preferred to change lanes before reaching the on-ramp

area. Therefore, more accidents would happen due to frequent speed deceleration and lane-changing rates.

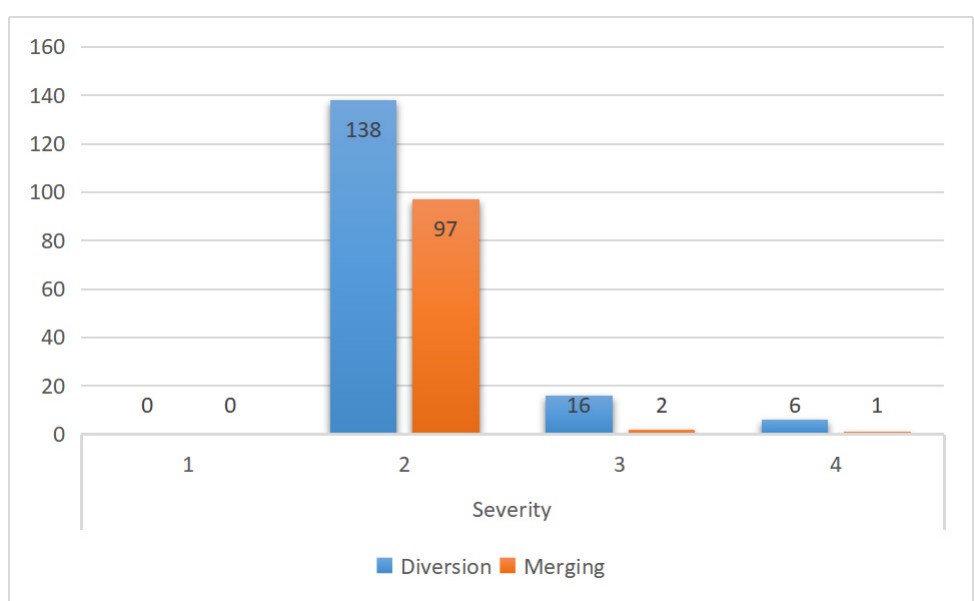

**Figure 3.** The statistical distribution of the parameter "Severity".

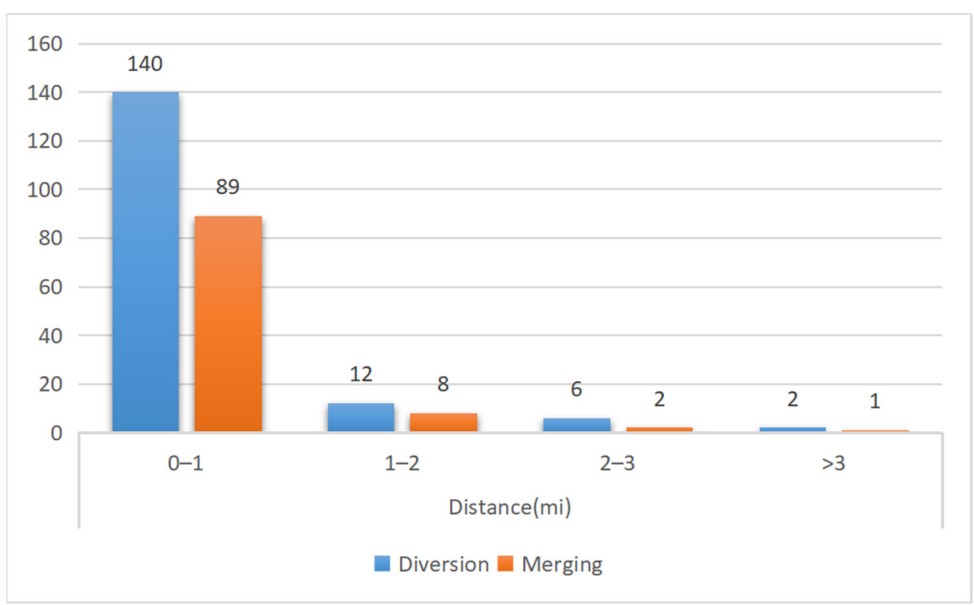

**Figure 4.** The statistical distribution of the parameter "Distance".

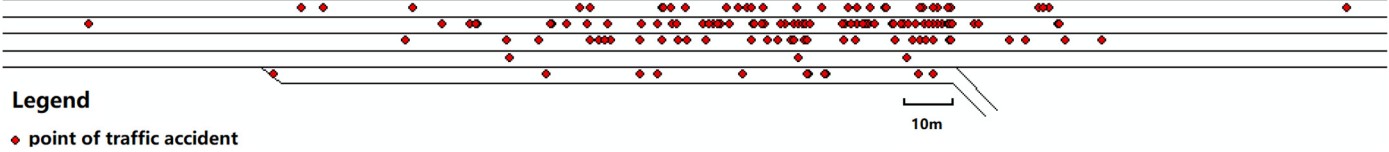

**Figure 5.** Spatial distribution of traffic accidents in the diversion area of the highway.

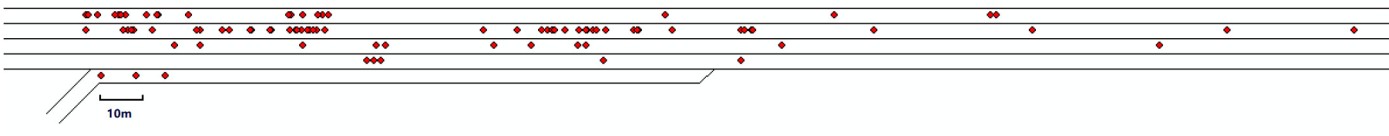

**Legend**

◆ point of traffic accident

**Figure 6.** Spatial distribution map of traffic accidents in the merging area of the highway.

## 4. Results

### 4.1. Spatial Autocorrelation Analysis

If the road conditions in the highway merging or diversion sections are the same at every point, and the traffic volume and vehicle types of each lane satisfy an ideal uniform distribution, it can be assumed that the location of the occurrence of an accident is independent of the lane and distance from the ramp. Based on this assumption, when the dataset is large enough, the traffic accidents should be evenly distributed. The accidents in a specific lane or area should not be significantly more than those in other places. Then, all accidents in the entire region can be considered as one category. However, in reality, the road conditions of each lane and section in the merging and diversion sections are different. Even in the same area, the traffic volume of each lane is different. The proportion of vehicles on the main road is significantly higher than that of vehicles on the off-ramp. The traffic volume of the first and second lanes is significantly higher than that of the third, fourth, and acceleration/deceleration lanes. Therefore, there are significant spatial differences in the occurrence of traffic accidents. It can be determined that the areas where a small number of accidents occur or no accidents occur account for most of the entire spatial area. Only a few areas will have accident hotspots. The more events occur in an area, the smaller the proportion it occupies in the overall section. Due to the uneven distribution of accidents, the clustering effect will gradually improve as the clustering radius increases from zero. The accident points with close spatial relationships are clustered into one category. However, when the threshold of the clustering radius is reached, with the increase in the clustering radius, the accident points with small correlation are also clustered, which reduces the clustering accuracy.

Therefore, before conducting the kernel density analysis, it is crucial to confirm that the spatial distribution of traffic accident points in the merging and diversion sections is clustered rather than randomly distributed. This study uses the "average nearest neighbor" parameter in ArcGIS to measure the spatial autocorrelation degree based on the feature's location. The similarity between the average distance and the assumed random distribution distance is measured according to the distance between each feature and its nearest neighbor. Then, the Z-score is returned. If the Z-score is negative and the *p*-value is small, the distribution of the dataset tends to converge, and vice versa. Figure 7 shows the results of the statistical significance in the merging and diversion sections.

According to Figure 7, the Z-values for the diversion area and the merging area are −3.79 and −5.06, respectively, and both *p*-values are less than 0.05. This means that the observed spatial patterns are unlikely to result from random processes (very low probability), so the null hypothesis can be rejected. As shown in Figure 7, the confidence levels are both greater than 90%, concluding that the distribution of traffic accidents in the highway diversion and merging areas follows a clustering distribution pattern, and the spatial clustering method based on kernel density can effectively reflect the micro-distribution patterns of accident points in the diversion and merging areas, which validates the applicability of this method.

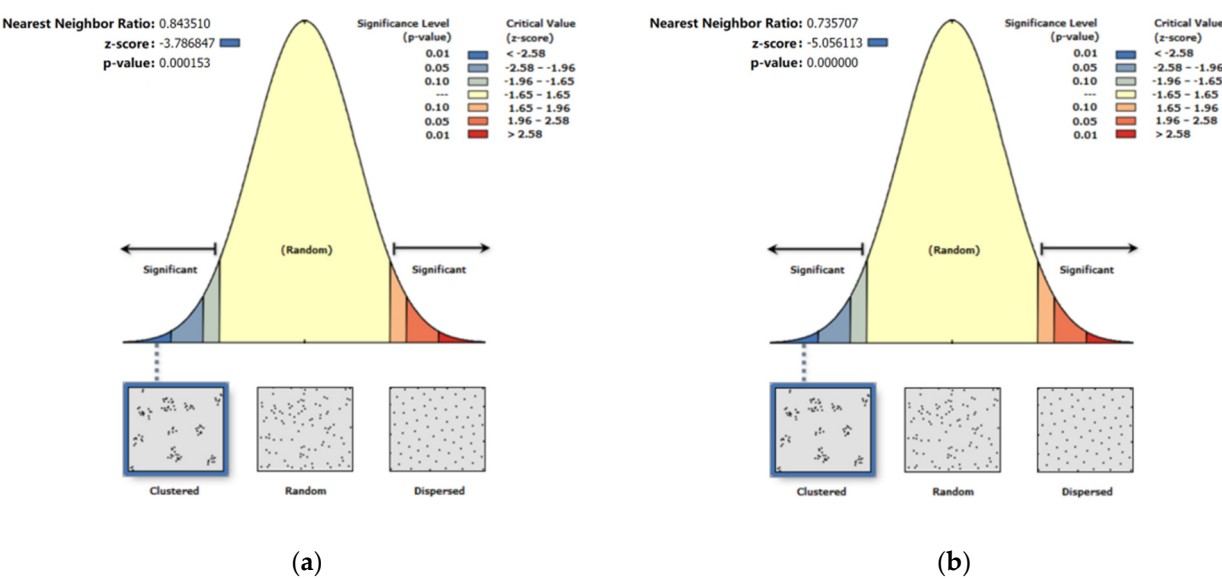

**Figure 7.** Average nearest neighbor statistical significance results. (**a**) Diversion area; (**b**) merging area.

### 4.2. Kernel Density Analysis

Firstly, according to the method proposed in the previous section, the accident location data in the highway diversion and merging areas were extracted. These data were marked on the standard highway section, as shown in Figures 5 and 6. Secondly, the kernel density method was applied to create a traffic accident density map. Since the window width h will affect the smoothness of the density function estimation and the density estimation of sample points for x, it is crucial to choose an appropriate width value. Generally, as the window width h increases, the spatial point density changes will become smoother. However, the point density changes will fluctuate greatly when the window width h decreases. Therefore, the relative "peaks" of low-density areas can be determined to identify the clustering range. In this study, the "Silverman empirical rule" spatial variable was used to calculate the default search radius (bandwidth) specifically for the input dataset. This can effectively avoid spatial outliers (points that are too far away from other points). Based on this, the kernel density analysis results were classified into four categories (Area 1 to Area 4), and the kernel density analysis results of the diversion and merging areas were obtained, shown in Figure 8.

### 4.3. Accident Cause Analysis of Four-Level Accident-Prone Areas

#### 4.3.1. Multinomial Logit Regression Model

To determine the possible factors that contribute to the spatial distribution differences of accident positions, this study uses the logistic regression model for analysis.

The logistic regression model mainly studies the dependency relationship between the dependent variable and the independent variables. It requires the dependent variable to be a categorical variable (binary or multicategory). The independent variables can be continuous variables, ordinal variables, or categorical variables. However, in practical problems, many variables are not continuous variables or ordinal variables. The variables of the traffic accident distribution area have unordered and multivariate attributes. Therefore, an unordered multicategory multivariate logistic regression model is suitable. This model first defines one level of the dependent variable as the reference level. In this study, the area with the smallest value in the kernel density analysis results is selected as the reference

group, and the correlation between the other three areas and the independent variables is analyzed. Three generalized logistic models are constructed, as shown in Equations (5)–(7):

$$\ln(P_1/P_4) = \alpha_1 + \sum_{k=1}^{k} \beta_{1k}x_k \tag{5}$$

$$\ln(P_2/P_4) = \alpha_2 + \sum_{k=1}^{k} \beta_{2k}x_k \tag{6}$$

$$\ln(P_3/P_4) = \alpha_3 + \sum_{k=1}^{k} \beta_{3k}x_k \tag{7}$$

where $\alpha_n$ is the constant term, $x_k$ is the explanatory variable representing the kth influencing factor, and $\beta_{nk}$ is the regression coefficient of the kth influencing factor in model n.

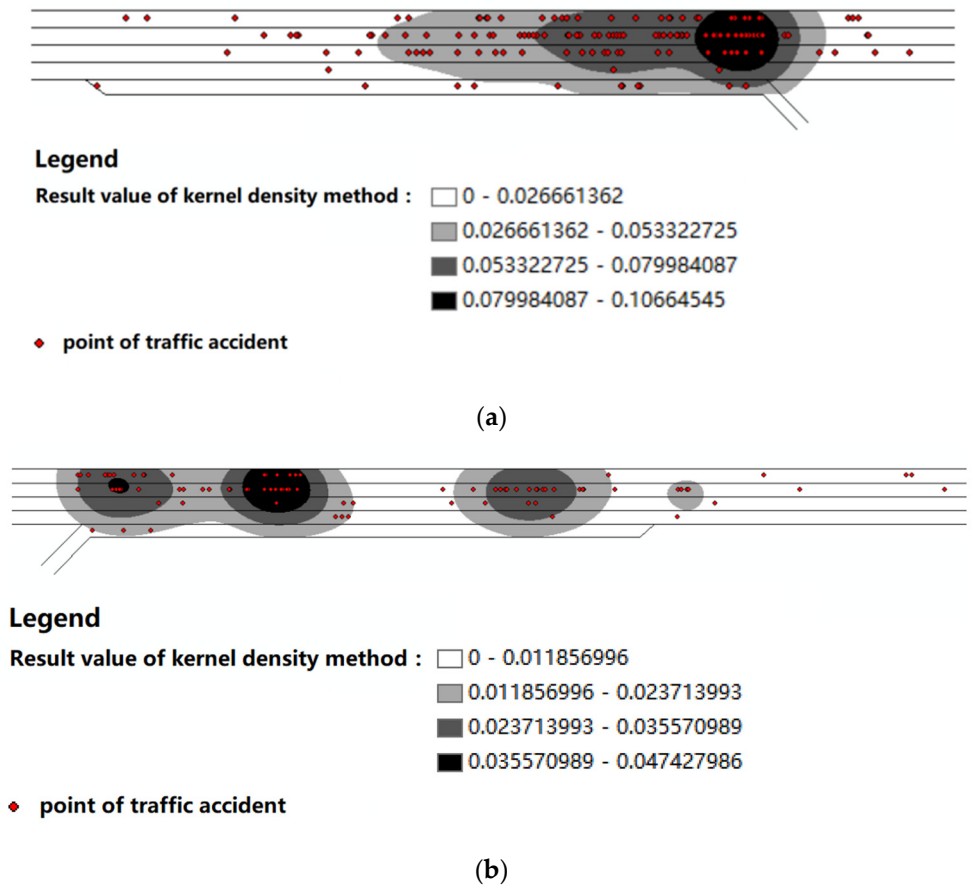

**Figure 8.** Kernel density analysis result of diversion area and merging area. (**a**) Diversion area; (**b**) merging area.

This study used MLE (maximum likelihood estimation) to estimate parameters a and b. The estimated values of parameters $\alpha$ and $\beta$ can be obtained from the following likelihood equations shown in Equations (8) and (9):

$$\frac{\partial LL(\alpha, \beta_K)}{\partial \alpha} = \sum_{i=1}^{N} \left[ y_i - \frac{e^{(\alpha + \beta_K x_{ki})}}{1 + e^{(\alpha + \beta_K x_{ki})}} \right] = 0 \tag{8}$$

$$\frac{\partial LL(\alpha, \beta_K)}{\partial \beta_K} = \sum_{i=1}^{N} \left[ y_i - \frac{e^{(\alpha + \beta_K x_{ki})}}{1 + e^{(\alpha + \beta_K x_{ki})}} \right] x_{ki} = 0 \tag{9}$$

By following the above steps, an initial model was obtained, but not every factor in the model was appropriate. In this case, an optimized model was obtained using AIC (Akaike information criterion). AIC is a concrete method of the principle of parsimony. The smaller the AIC value, the better the model. In each step, the least important factor was removed. Then, the model was gradually refined until the AIC reached the minimum value.

There are various factors that affect the spatial distribution of accidents, such as time, space, and other external factors. Therefore, based on the existing literature and the original fields contained in the accident database, this study used seven explanatory variables to explain the spatial distribution of accidents, as shown in Table 2. Due to the excessively concentrated distribution of the "Severity" variable, using it as an independent input in the model may lead to overfitting and inaccurate results. Therefore, this article did not take into account the severity of accidents in the causal analysis.

**Table 2.** Variable explanation for accident data.

| Variable Name | Variable Type | Variable Classification | Value of Dummy Variable |
|---|---|---|---|
| Dependent variable | | | |
| Area where the accident took place | Classification variables | Level 1 area | - |
| | | Level 2 area | - |
| | | Level 3 area | - |
| | | Level 4 area * | - |
| Independent variable | | | |
| Weather condition | Classification variable | Sunny | 10 |
| | | Cloudy | 01 |
| | | Rain and Snow * | 00 |
| Time | Classification variable | Day | 1 |
| | | Night * | 0 |
| Lane | Classification variable | Lane 1 | 1000 |
| | | Lane 2 | 0100 |
| | | Lane 3 | 0010 |
| | | Lane 4 | 0001 |
| | | Acceleration or deceleration lanes * | 0000 |
| Longitudinal distance from the ramp (m) | Continuous variable | - | - |
| Humidity (%) | Continuous variable | - | - |
| Temperature (F) | Continuous variable | - | - |
| Visibility (mi) | Continuous variable | - | - |

Note: * is the control group in the categorical variable group.

When analyzing the correlation of traffic accident spatial distribution differences, the data need to be processed before building the model. The data in various fields of the accident database were discretized, and the results are shown in Tables 3 and 4, which summarize the statistical results of accident data in the diversion and merging areas.

In the diversion area, accidents occurring in the secondary and primary areas account for large proportions of 32.5% and 26.9%, respectively. In the merging area, accidents occurring in the four areas are distributed quite evenly from the primary area to the fourth area, with proportions of 25%, 28%, 26%, and 21%, respectively. As for weather conditions, the statistical distribution of the diversion area and merging area is also similar, with approximately half of the total accidents occurring on clear days and non-clear days (cloudy and rainy/snowy). In terms of the time of day, accidents in the diversion area occur more during the daytime (83.75%), while in the merging area, the proportion of accidents during the daytime is only 58%. Regardless of whether they are in the diversion

area or merging area, accidents occurring in the second lane are of the highest proportions of 45.625% and 57%, respectively. In the diversion area, accidents occurring in the fourth lane are the least frequent, accounting for only 1.875%, while in the merging area, accidents occurring in the acceleration lane are the least frequent, accounting for only 3%. As for the longitudinal distribution of accidents, in the diversion area, the farthest accident point is 193.0 m away from the ramp, while in the merging area, the farthest accident point can reach up to 447.9 m. As for the three variables of humidity, temperature, and visibility, the data in the diversion area and merging area are statistically similar.

**Table 3.** Statistical results of discrete variables.

| Area | Variable Name | Variable Classification | Counts | Percentage |
|------|---------------|------------------------|--------|------------|
| Diversion area | Area where the accident took place | Level 1 area | 43 | 26.9% |
| | | Level 2 area | 52 | 32.5% |
| | | Level 3 area | 36 | 22.5% |
| | | Level 4 area * | 29 | 18.1% |
| | Weather condition | Sunny | 80 | 50% |
| | | Cloudy | 65 | 40.6% |
| | | Rain and Snow * | 15 | 9.4% |
| | Time | Day | 134 | 83.8% |
| | | Night * | 26 | 16.2% |
| | Lane | Lane 1 | 34 | 21.2% |
| | | Lane 2 | 73 | 45.6% |
| | | Lane 3 | 38 | 23.8% |
| | | Lane 4 | 3 | 1.9% |
| | | Deceleration lane * | 12 | 7.5% |
| Merging area | Area where the accident took place | Level 1 area | 25 | 25.0% |
| | | Level 2 area | 28 | 28.0% |
| | | Level 3 area | 26 | 26.0% |
| | | Level 4 area * | 21 | 21.0% |
| | Weather condition | Sunny | 45 | 45.0% |
| | | Cloudy | 39 | 39.0% |
| | | Rain and Snow * | 16 | 16.0% |
| | Time | Day | 58 | 58.0% |
| | | Night * | 42 | 42.0% |
| | Lane | Lane 1 | 21 | 21.0% |
| | | Lane 2 | 57 | 57.0% |
| | | Lane 3 | 13 | 13.0% |
| | | Lane 4 | 6 | 6.0% |
| | | Acceleration lane * | 3 | 3.0% |

Note: * is the control group in the categorical variable group.

**Table 4.** Statistical results of continuous variables.

| Area | Variable Name | Minimum Value | Maximum Value | Mean | Standard Deviation |
|------|---------------|---------------|---------------|------|--------------------|
| Diversion area | Longitudinal distance from the ramp (negative value represents the upstream distance from the ramp) | −193.0 | 88.2 | −35.2 | 40.1 |
| | Humidity (%) | 6.0 | 96.0 | 56.2 | 22.7 |
| | Temperature (F) | 10.4 | 91.4 | 65.1 | 12.9 |
| | Visibility (mi) | 0.2 | 10.0 | 9.2 | 2.2 |
| Merging area | Longitudinal distance from the ramp | 0.2 | 447.9 | 104.3 | 102.4 |
| | Humidity (%) | 15.0 | 93.0 | 63.5 | 19.5 |
| | Temperature (F) | 9.0 | 100.0 | 58.1 | 15.4 |
| | Visibility (mi) | 1.0 | 10.0 | 8.5 | 2.7 |

### 4.3.2. Model Results

This study used multiple unordered logistic regression to analyze 160 accidents in the diversion area and 100 accidents in the merging area. The dependent variable was divided into four categories (Area 1 to Area 4) based on the values from the kernel density analysis. The initial model was optimized based on AIC. In each step, an unimportant factor was removed until the AIC reached the minimum value. The odds ratio (OR) was used to evaluate the impact of a unit change in each independent or explanatory variable on the ratio. Estimating the OR of each variable in the model helps to discover the underlying relationship between the accident distribution and variables. The positive or negative influence of a variable can be expressed as OR > 1 or OR < 1. In addition, OR = 1 indicates that the factor has no significant impact on accidents. The estimated coefficients, standard errors, and significance of each variable in the model are presented in Tables 5 and 6. The stepwise process of AIC is shown in Tables 7 and 8.

**Table 5.** Estimated results of the multinomial logit model for the diversion areas.

| Model | Variable | Coefficient Estimates | Estimated Standard Error | Significance | OR Value |
|---|---|---|---|---|---|
| Model 1 | Intercept | −4.145 | 3.481 | 0.234 | |
| | Longitudinal distance from the ramp | 0.081 | 0.014 | 0.000 | 1.084 |
| | Humidity (%) | 0.033 | 0.016 | 0.040 | 1.034 |
| | Visibility | 0.519 | 0.289 | 0.073 | 1.680 |
| | Occur in lane 1—No | 0.864 | 0.863 | 0.316 | 2.373 |
| | Occur in lane 1—Yes | 0 * | | | |
| | Occur in lane 2—No | −1.568 | 0.721 | 0.030 | 0.208 |
| | Occur in lane 2—Yes | 0 * | | | |
| Model 2 | Intercept | 1.979 | 1.886 | 0.294 | |
| | Longitudinal distance from the ramp | 0.032 | 0.008 | 0.000 | 1.033 |
| | Humidity (%) | 0.014 | 0.013 | 0.267 | 1.015 |
| | Visibility | −0.007 | 0.130 | 0.956 | 0.993 |
| | Occur in lane 1—No | −0.017 | 0.669 | 0.980 | 0.983 |
| | Occur in lane 1—Yes | 0 * | | | |
| | Occur in lane 2—No | −1.176 | 0.620 | 0.058 | 0.309 |
| | Occur in lane 2—Yes | 0 * | | | |
| Model 3 | Intercept | 0.994 | 1.779 | 0.576 | |
| | Longitudinal distance from the ramp | 0.006 | 0.006 | 0.391 | 1.006 |
| | Humidity (%) | −0.003 | 0.013 | 0.802 | 0.997 |
| | Visibility | −0.101 | 0.121 | 0.402 | 0.904 |
| | Occur in lane 1—No | 0.723 | 0.645 | 0.262 | 2.061 |
| | Occur in lane 1—Yes | 0 * | | | |
| | Occur in lane 2—No | 0.140 | 0.596 | 0.814 | 1.150 |
| | Occur in lane 2—Yes | 0 * | | | |

Note: * is the control group in the categorical variable group.

The coefficient of determination ($R^2$), mean squared error (*MSE*), and mean absolute error (*MAE*) were used to evaluate the accuracy of the model (see Equations (10)–(12)).

$$R^2 = 1 - \frac{\sum_{i=1}^{n}(y_i - \hat{y}_i)^2}{\sum_{i=1}^{n}(y_i - \overline{y}_i)^2} \tag{10}$$

$$MSE = \frac{1}{n}\sum_{i=1}^{n}(\hat{y}_i - y_i)^2 \tag{11}$$

$$MAE = \frac{1}{n}\sum_{i=1}^{n}|\hat{y}_i - y_i| \tag{12}$$

where $n$ is the number of samples, $y_i$ is the true value, $\hat{y}_i$ is the predicted value, and $\overline{y}_i$ is the average value. *MSE* and *MAE* are often used in regression analysis to measure the difference between predicted and actual values.

Table 9 shows that compared with the ordered logit model, the multinomial logit model has better performance and predictive accuracy on the dataset. The multinomial logit model has a relatively high $R^2$ value and low *MAE* and *MSE* values. Therefore, the multinomial logit model is more suitable for predicting the four levels of accident-prone areas. The ordered logit model may be limited when dealing with unordered variables because it requires the order or hierarchy of dependent variables to be determined before modeling. In addition, the multinomial logit model can better capture the nonlinear relationship and interaction effects of the dependent variable, which improves the prediction accuracy.

**Table 6.** Estimated results of the multinomial logit model for the merging areas.

| | Variable | Coefficient Estimates | Estimated Standard Error | Significance | OR Value |
|---|---|---|---|---|---|
| Model 1 | Intercept | 5.940 | 4.157 | 0.153 | |
| | Longitudinal distance from the ramp | −0.086 | 0.023 | 0.000 | 0.918 |
| | Temperature | 0.190 | 0.069 | 0.006 | 1.210 |
| | Daytime—No | 2.689 | 1.516 | 0.076 | 14.710 |
| | Daytime—Yes | 0 * | | | |
| | Occur in lane 1—No | −5.414 | 3.269 | 0.098 | 0.004 |
| | Occur in lane 1—Yes | 0 * | | | |
| | Occur in lane 2—No | −10.721 | 2.941 | 0.000 | $2.207 \times 10^{-5}$ |
| | Occur in lane 2—Yes | 0 * | | | |
| | Sunny—No | 3.313 | 1.663 | 0.046 | 27.467 |
| | Sunny—Yes | 0 * | | | |
| Model 2 | Intercept | 2.568 | 3.878 | 0.508 | |
| | Longitudinal distance from the ramp | −0.071 | 0.022 | 0.001 | 0.932 |
| | Temperature | 0.205 | 0.069 | 0.003 | 1.228 |
| | Daytime—No | 2.347 | 1.484 | 0.114 | 10.450 |
| | Daytime—Yes | 0 * | | | |
| | Occur in lane 1—No | −3.741 | 2.994 | 0.211 | 0.024 |
| | Occur in lane 1—Yes | 0 * | | | |
| | Occur in lane 2—No | −7.845 | 2.584 | 0.002 | 0.000 |
| | Occur in lane 2—Yes | 0 * | | | |
| | Sunny—No | 2.751 | 1.627 | 0.091 | 15.655 |
| | Sunny—Yes | 0 * | | | |
| Model 3 | Intercept | 0.357 | 3.702 | 0.923 | |
| | Longitudinal distance from the ramp | −0.062 | 0.022 | 0.005 | 0.940 |
| | Temperature | 0.177 | 0.067 | 0.008 | 1.194 |
| | Daytime—No | 2.439 | 1.401 | 0.082 | 11.462 |
| | Daytime—Yes | 0 * | | | |
| | Occur in lane 1—No | −1.170 | 2.971 | 0.694 | 0.310 |
| | Occur in lane 1—Yes | 0 * | | | |
| | Occur in lane 2—No | −6.036 | 2.482 | 0.015 | 0.002 |
| | Occur in lane 2—Yes | 0 * | | | |
| | Sunny—No | 3.182 | 1.562 | 0.042 | 24.083 |
| | Sunny—Yes | 0 * | | | |

Note: * is the control group in the categorical variable group.

**Table 7.** The stepwise process of AIC in the diversion areas.

| Steps | Deleted Variable | Variable Significance | AIC Value |
|---|---|---|---|
| Initial | / | / | 389.071 |
| 1 | Sunny | 0.986 | 383.213 |
| 2 | Temperature | 0.864 | 377.952 |
| 3 | Daytime | 0.697 | 373.388 |
| 4 | Cloudy | 0.402 | 370.322 |
| 5 | Lane 4 | 0.157 | 369.527 |
| 6 | Lane 3 | 0.033 | 372.258 |

**Table 8.** The stepwise process of AIC in the merging areas.

| Steps | Deleted Variable | Variable Significance | AIC Value |
|---|---|---|---|
| Initial | / | / | 225.114 |
| 1 | Cloudy | 0.703 | 220.525 |
| 2 | Visibility | 0.822 | 215.438 |
| 3 | Humidity | 0.390 | 212.449 |
| 4 | Lane 3 | 0.298 | 210.130 |
| 5 | Lane 4 | 0.068 | 207.565 |

**Table 9.** The cross-validation results of the factors affecting accident occurrence.

| Model | Area | $R^2$ | *MSE* | *MAE* |
|---|---|---|---|---|
| Multinomial Logit Model | Diversion | 0.921 | 0.091 | 0.083 |
| | Merging | 0.865 | 0.125 | 0.144 |
| Ordered Logit Model | Diversion | 0.652 | 0.267 | 0.375 |
| | Merging | 0.804 | 0.222 | 0.342 |

## 5. Discussion

### 5.1. Analysis of Influencing Factors in Different Accident-Prone Areas in the Diversion Area

The model results show that for the diversion area, the explanatory variables "longitudinal distance from the ramp", "humidity", "visibility", "lane 1", and "lane 2" have relatively significant effects on the model. In Area 1, "longitudinal distance from the ramp", "humidity", and "lane 2" have significant effects on accidents in Area 1. Among them, the longitudinal distance from the ramp and humidity have positive effects. This indicates that compared with Area 4, especially in the upstream area, the closer the distance from the ramp, the greater the possibility of accidents occurring in Area 1. The greater the humidity, the more likely accidents occur in Area 1. Compared with Area 4, the probability of accidents not occurring in lane 2 in Area 1 is low, which means most accidents in Area 1 occur in lane 2. For Area 2, only the variable "longitudinal distance from the ramp" has a significant effect, which is similar to Area 1. In the upstream area, the closer the distance to the ramp, the greater the possibility of accidents occurring in Area 2. Area 3 has no significant influential variables.

### 5.2. Analysis of Influencing Factors of Different Accident-Prone Areas in the Merging Zone

In the merging zone, the variables "distance from the nose in the longitudinal direction", "temperature", "daytime", "lane 1", "lane 2", and "clear weather" have relatively statistically significant effects on the model. From the results of the merging region model parameter estimation, it is found that in Area 1, variable 1 "longitudinal distance from exit", "temperature", "lane 2", and weather all have significant effects on whether the accident occurs in Area 1. Unlike in the diversion area, the longitudinal distance from the nose has a negative correlation with accidents occurring in Area 1; that is, the farther the longitudinal distance, the less likely it is to occur in Area 1, which is consistent with the kernel density result graph, where Area 1 of the merging zone is located far from the ramp. The higher the temperature, the more likely accidents will occur in Area 1. Compared with Area 4, the probability of accidents occurring in lane 2 is higher in Area 1. When the weather is not good (not clear weather), accidents are more likely to occur in Area 1. In Area 2, the pattern of accidents is similar to that in Area 1, but more accidents occur at night. Area 3 also has roughly the same pattern: accidents are more likely to occur in the second lane when the weather is not good and the temperature is higher.

### 5.3. Comparative Analysis of Factors Affecting Accident Occurrence in the Merging and Diversion Areas

By comparing the accident occurrence patterns in the merging and diversion areas, the following patterns can be observed. In the diversion area, there is a significant positive correlation between the area distribution of accident occurrence locations and the longitudinal distance from the ramp. This means that the farther the accident is from the upstream, the lower the accident density in the accident area. On the other hand, there is no obvious longitudinal distribution pattern in the accident area distribution in the merging area. Humidity has a certain influence on the distribution of accidents in the diversion area. However, temperature and weather have a significant impact on the distribution of accidents in the merging area. Regardless of whether it is in the diversion or merging area, the accident rate in the second lane is the highest. Based on the analysis results, the following conclusions can be drawn.

a. The accident-prone areas in the diversion area are more concentrated than those in the merging area. The accident density level distribution is closer to the cross-section near the ramp. In contrast, there is often more than one accident-prone area in the merging area. This is because, in the high-speed diversion area (500 m before and after the ramp), some drivers who are not familiar with the road conditions will suddenly slow down near the ramp. If the vehicle behind them does not slow down in time, rear-end accidents will inevitably occur.

b. In the merging area, accidents are more spread out than in the diversion zone. This is because vehicles are generally in the acceleration lane before entering the main road from the ramp and need to accelerate to a specific speed to enter the main road. The acceleration of vehicles in the acceleration lane varies widely, so the location where vehicles accelerate to the speed limit and then change lanes to enter the main road varies. Therefore, accidents in the merging zone appear to be more dispersed than accidents in the diversion zone.

c. In the diversion areas and merging areas, accidents occurring in the second lane account for the highest proportion among all lanes. This is because the second lane is the lane with the most vehicles, and the probability of accidents occurring is relatively high. Moreover, if the vehicles in the overtaking lane (always travel fast) need to get off the highway when approaching the ramp, they need to change lanes, which can easily lead to side collisions and rear-end collisions with adjacent vehicles. Other lanes have relatively fewer vehicles near the ramp, so accidents are more likely to occur in the second lane.

d. The model shows that rain increases the probability of vehicle accidents, so necessary traffic control and traffic warning measures should be implemented in the merging and diversion areas on rainy days.

e. The model indicates that the accident rate at night is much higher than that during the day, which is also mentioned in the research of Chen et al. [44]. This is mainly because the driver's visibility is poor at night, and there are almost no lighting facilities on the highway except in service areas and tunnels. Drivers can only judge the road conditions ahead based on the vehicle lights. If they are driving on a complex road section (such as the merging and diversion areas) or face a sudden situation, the driver cannot make a prediction in advance, which can easily lead to the occurrence of traffic accidents. This is consistent with the study results of Wei et al. [45].

## 6. Conclusions

The main contribution of this paper is identifying accident-prone areas near highway ramps using spatial autocorrelation analysis from the microscopic aspect. This approach differs from traditional methods that rely on a statistical analysis of accident data without considering the spatial relationships between accidents. It classified the accident-prone areas into four levels with specific spatial division results. The main differences between the accident distribution and causes were presented according to the analysis and model results. The main conclusions can be drawn as follows.

Firstly, based on kernel density analysis of accident data in highway diversion and merging areas, this study found that the accident-prone locations were mainly located at the highway entrances and exits. This is consistent with normal expectations because these locations are usually areas with high traffic flow, fast speeds, and frequent lane-changing, which are the main factors of accidents. Secondly, when analyzing the time period and weather conditions of different accident-prone areas, it was found that accidents occurred more often in poor weather conditions and at night. This indicates that drivers need to pay attention to driving safety in such situations, especially near the highway ramp (within 100 m). In this study, spatial clustering analysis of accident-prone points was conducted. A significant spatial correlation was found among the high accident-prone points. These points were usually located in specific areas near the ramp, which need more management to reduce the accident rates. Finally, a multinomial logit model was used to analyze the causes of spatial differences in accident distribution. It was found that temperature, the accident lane, weather, and the accident time were important factors

affecting the spatial distribution of traffic accidents. According to these findings, this study suggests that different preventive measures should be used for different types of traffic accidents. For example, the flexible control of speed limits, THW, warning signs, and lane markings should be strengthened in the highway diversion area to improve drivers' safety awareness. In summary, the results of this study provide an important decision basis for traffic management departments to adopt a refined management strategy for the diversion and merging areas to reduce the accident rate.

In addition to the above results, this paper has the following shortcomings. First, this study only analyzed the accident location, time, and weather, and ignored the influence of different levels of accident areas on drivers' subjective behaviors in depth. Second, this study only used the U.S. highway accident dataset, which to some extent limits the generalizability of the conclusions. Third, the proposed management measures have not been validated in realistic scenarios.

To address the above problems, a combination of field experiments and simulations should be realized in future research. This will help to comprehensively compare and analyze the differences between accidents near the ramp and accidents on normal sections. The kernel density algorithm can also be combined with artificial intelligence technology to analyze the implied relationships in traffic accidents to obtain more accurate accident causation analysis results. Moreover, different traffic rules can be considered as the causative factors of accidents to provide a scientific basis for improving the management of ramps.

**Author Contributions:** Conceptualization, Q.Y.; Data curation, W.S.; Funding acquisition, Y.L.; Investigation, Q.Y.; Methodology, Y.L.; Software, W.S.; Validation, Z.X.; Writing—original draft, Z.X. All authors have read and agreed to the published version of the manuscript.

**Funding:** This research was funded by the Fund of National Engineering and Research Center for Mountainous Highways, grant number GSGZJ-2022-09. This research was also funded by the National Natural Science Foundation of China, grant number 52202419.

**Data Availability Statement:** The data that support the findings of this study are available from the corresponding author upon reasonable request.

**Conflicts of Interest:** The authors declare no conflict of interest.

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
