# Peer review of "Division and Analysis of Accident-Prone Areas near Highway Ramps Based on Spatial Autocorrelation"

_sustainability, doi:10.3390/su15107942_

Round 1
Reviewer 1 Report
The research conducted is interesting and overall well done. However, I provide some suggestions to improve it, also concerning the readability of the paper.
1) literature review: automatic vehicles are mentioned. it could be useful to write a few lines about the problems that could bring this type of vehicle into the studied area (ramp).
2) highlight more the objective and what the study conducted introduces into the technical/scientific literature.
3) the flowchart is a little contextualized. Perhaps it could be moved to a paragraph that provides an overview of the overall research. the other paragraphs of the methodology should then resume and illustrate in detail what is briefly described in this phase.
4) equation 1: format problem
5) equation 3 and the following: improve the format
6) improve the description of the database. information about the data included in the database and the elementary statistic could be described.
7) the par. 3.2 seems to be incomplete. After the pictures maybe the suggested information could be added.
8) methodology could be described in order to replicate the research. Therefore, more detailed information concerning the steps conducted needs to be written. Furthermore, the methodology paragraphs should be the same as the flowchart blocks.
9) it would be nice to read some considerations on the technical interpretation of the results obtained. are consistent with the existing literature. what am I not? what does this come from? what do they mean technically?
10) 4.3.1. the paragraph mixes methodology and results
11) a factor analysis could help the interpretation of the results.
Reviewer 2 Report
I doubt the novelty and contribution of this paper as there are some research that tackled similar issues and were not referenced such as “Understand the impact of traffic states on crash risk in the vicinities of Type A weaving segments: A deep learning approach”
Also, the authors should shed some light on the Highway Capacity Manual methodology when dealing with merging and diverging situations and how the manual define the weaving area/ramp influence area.
The severity of the collisions should be explored/included in the model.
There should be some discussion about the collisions in 2020 and 2021 and how it was impacted by the COVID and the lockdowns and whether these lockdowns impacted the results.
Reviewer 3 Report
This paper is very interesting, well structured, and seeks to provide a solution on identifying accident-prone areas near the highway ramps, including diversion areas and merging areas.
A thorough revision and editing is required for checking typos and sentence structure. Also, the following will need to be addressed.
i. i) Equation 1 is not clear.
ii. ii)Line 43 needs to be supported with evidence from literature.
The authors should consider adding literature on the use of machine learning algorithms for the Division and Analysis of Accident-prone such as:
Patil, J., Prabhu, M., Walavalkar, D. and Lobo, V.B., 2020, December. Road accident analysis using machine learning. In 2020 IEEE Pune Section International Conference (PuneCon) (pp. 108-112). IEEE.
Hasib, K.M., Showrov, M.I.H. and Das, A., 2020, September. Accidental prone area detection in bangladesh using machine learning model. In 2020 3rd International Conference on Computer and Informatics Engineering (IC2IE) (pp. 58-62). IEEE.
Vitianingsih, A.V., Suryana, N. and Othman, Z., 2021. Spatial analysis model for traffic accident-prone roads classification: A proposed framework. IAES International Journal of Artificial Intelligence, 10(2), p.365.
Sumanth, P., Anudeep, P.S. and Divya, S., 2020. Analysis of machine learning algorithm with road accidents data sets. International Journal of Engineering and Management Research, e-ISSN, pp.2250-0758.
Finally, the authors should better highlight the methodological contribution with respect to the state of the art: in fact, it seems that mentioned literature (such as [29]; [30]; [31]) could be used to solve the same problem, thus limiting the novelty and contribution of the paper. The evaluation should compare the proposed methodology with the state of the art, if possible.
Round 2
Reviewer 1 Report
Thank you for addressing the reviewer's comments.
Author Response
The reviewer did not present further comments.
Reviewer 2 Report
.
Author Response

(The authors gave the same response as above.)

Reviewer 3 Report
On the second submission, the authors have addressed the critical comments. The authors should also provide a cross validation of the factors affecting accident occurrence. This shall provide a comparative assessment with existing literature
